# Physiological Characterization of Drought Responses and Screening of Rice Varieties under Dry Cultivation

**Xiaoshuang Wei** [1,†], **Baifeng Cang** [1,†], **Kuo Yu** [1], **Wanchun Li** [1], **Ping Tian** [1], **Xiao Han** [1], **Guan Wang** [1], **Yuting Di** [1], **Zhihai Wu** [1,*] and **Meiying Yang** [2]

[1] College of Agronomy, Jilin Agricultural University, National Characteristic Station for Crop Variety Approval, Changchun 130118, China

[2] College of Life Sciences, Jilin Agricultural University, Changchun 130118, China

[*] Correspondence: wuzhihai1116@163.com

[†] These authors contributed equally to this work.

**Abstract:** Drought is a serious factor limiting rice production, and it leads to huge economic losses. Considering the current and projected global food demand, increasing productivity of drought-prone crops has become critical. In order to achieve the production target, rice drought-tolerant germplasm resources are an important prerequisite for the development of water-saving cultivation. Through multi-indicator measurement, the stress effect of drought on rice was clarified and a preliminary drought resistance identification index system was established based on the response of plant the germination, seedling and adult stages of rice and materials suitable for dry cultivation were screened. The results showed that relative root length, relative root weight and relative shoot weight were most affected by drought stress at the germination stage, while root length and root dry weight were positively correlated with the drought survival proportion at the seedling stage; high net photosynthetic rates and antioxidant enzyme activities are maintained in the late period in strongly drought-tolerant varieties. In this experiment, two drought-resistant varieties were screened, there was a high consistency in the screening of drought-tolerant varieties at the germination and seedling stages, with their joint screening showing the same performance as at the adult stage. The drought-resistant varieties at the adult stage can promote seed filling and ensure group yield by prolonging photosynthesis time and enhancing antioxidant enzyme activity, which can provide theoretical support and material basis for future variety screening and evaluation, as well as rice dry-crop cultivation.

**Keywords:** rice; dry cultivation; photosynthesis; antioxidant enzymes; yield; variety screening

## 1. Introduction

As a "semi-aquatic plant", rice (*Oryza sativa* L.) is primarily raised in water. With the decrease of freshwater resources and the increase of water consumption in agriculture, the shortage of water resources has become a major cause of rice production [1]. China's agriculture accounts for the largest proportion of the world's water use, with rice irrigation accounting for approximately 54% of the country's total water use and 62.5% of total agricultural irrigation water use [2]. Therefore, the traditional hydroponic cultivation of rice is gradually being transformed into water-saving cultivation modes, such as Dry-Wet cycling, wetting irrigation techniques and plastic film mulching cultivation techniques [3–5]. Rice under dry cultivation is a kind of rice crop that is directly sown under dry land conditions and primarily relies on natural precipitation during the whole reproductive period, with appropriate recharge of water only during critical periods of water demand or in times of drought [5]. In the context of the development of water-saving cultivation, rice under dry cultivation has become one of the main development directions of water-saving cultivation at present. In addition, the physiological and molecular mechanisms of drought-tolerant crops are not yet known. Therefore, a comprehensive understanding

of the mechanisms of drought tolerance is necessary to ensure security and sustainable agricultural development of energy and food crops.

Evaluating drought tolerance and the mechanism of drought resistance in crops is complex due to the genetic interaction with environmental factors [6,7]. Differences in the agronomic and physiological responses of rice caused by drought at different times are an important basis for evaluating the drought resistance of rice. Drought stress at the germination stage reduces rice shoot-related traits, root-related traits and germination rate, and each indicator is inhibited to different degrees [8,9]. The agronomic traits and physiological characteristics of different rice varieties are significantly affected by drought stress during the seedling stage, particularly in the cases of root and leaf development [10,11]. Photosynthesis is the main source of yield in rice [12], and the decrease in stomatal conductance and Mesophy II conductance under drought stress is the main reason for the decrease in photosynthetic rate [13], which significantly affects $CO_2$ assimilation. Crops facing drought stress activate the internal defense system to withstand the adverse growth conditions. For instance, antioxidant enzymes, such as peroxidase (POD), catalase (CAT), and superoxide dismutase (SOD), prevent reactive oxygen species (ROS) accumulation [14], and their levels are closely related to plant stress tolerance [15,16].

Selecting and breeding water-saving and drought-resistant rice varieties is an important prerequisite for promoting water-saving cultivation, and selecting and breeding drought-resistant varieties is a complex and tedious process. A large number of rice varieties have been evaluated for drought resistance at different stages, however, there are fewer varieties suitable for rice under dry cultivation, and previous research has predominantly evaluated them for a certain period, with poor representativeness, such as 'Yangda 6' (Indica) and 'Hanyou 8' (Japonica), which are only suitable for cultivation in subtropical monsoon climate regions [17]. Therefore, whole growing period screens for drought resistance is a necessary process for classifying varieties for resistance. However, due to increased heat resources and uneven rainfall distribution in the Northeast, there are no broad-spectrum dry-crop varieties of rice suitable for large-scale cultivation. In this experiment, 69 rice varieties were screened and identified for drought resistance throughout their growth period at three levels: germination stage, seedling stage and adult stage, with a view to screening varieties suitable for rice under dry cultivation in the Northeast China rice area and providing theoretical support for future variety selection, drought resistance identification and application.

## 2. Materials and Methods

### 2.1. Plant Materials and Treatment

This study was conducted at the National Characteristic Station for Crop Variety Approval at the Jilin Agricultural University in 2019–2021. Sixty-nine rice lines were selected in this experiment (Table S1).

### 2.1.1. Experiment I

The germination experiment was conducted using an artificial climate chamber. Fifty seeds of each variety were selected to be full and uniform, sterilized with 15% alcohol for 15 min, washed 3–5 times with distilled water, blotted dry with absorbent paper and placed in a Petri dish lined with 2 layers of filter paper for culture. In addition, 10 mL of 20% concentration polyethylene glycol (PEG-600 solution) was added to the treatment and 10 mL of distilled water was added to the control; each treatment was repeated 3 times. The artificial climate chamber is set up for 12 h of light incubation at 28 °C and 12 h of dark incubation at 22 °C, with 60% humidity.

### 2.1.2. Experiment II

The seedling drought screening experiments were carried out in pots. The pots were 17.5 cm in outer diameter, 14.6 cm in height, 12.5 cm in bottom diameter, and filled with 1.8 kg of soil. One-hundred seeds were sown in each pot by spot sowing, covered with

approximately 1 cm of soil, randomly arranged and replicated four times for each treatment. Fifty seedlings were retained at the 3-leaf stage by inter-planting. The first treatment began at the three-leafed stage, and when all varieties wilted at midday, 50% of varieties showed varying degrees of leaf wilting and a few varieties showed whole plant "dieback"; watering was carried out immediately, followed by a 120 h survey of seedling survival, after which watering was stopped and another drought treatment was carried out. When all varieties wilted again, and 50% of varieties showed varying degrees of whole plant 'dieback', a second watering was applied, followed by a 120 h survey of seedling survival.

### 2.1.3. Experiment III

The experiments were conducted by dry cultivation, with a hill spacing of 20 cm × 30 cm, with 8–12 seeds per hole. Each plot area was 5 m wide and 6 m long, and replicated three times in a randomized block design. The seeds were sown on 1 May and harvested on 12 October. They were irrigated by maintaining a water layer as a control, maintaining a moist nursery until the seedlings reached the three-leaf stage, maintaining a 2–3 cm water layer after the three-leaf stage, and drought stress as a treatment, with soil water potential maintained at −25 to −35 kPa (monitored by the soil water potential meter SYS-TSS1). Fertilizer was applied in a single application of calcium superphosphate ($P_2O_5$ 12%) 75 kg/ha and potassium dichromate ($K_2O$) 75 kg/ha as a base fertilizer and urea (N 46%) 160 kg/ha in batches of basic fertilizer: tiller fertilizer: panicle. fertilizer = 5:3:2. Where the soil organic matter was: 16.75 g/kg, available nitrogen: 160.53 mg/kg, available phosphorus:17.78 mg/kg and available potassium: 137.09 mg/kg.

### 2.2. Sampling and Measurement Methods
### 2.2.1. Germination Experiment

Measurements of the coleoptile length, shoot length, root length, number of roots, shoot weight, root weight and germination rate of each variety were counted on day 8 of culture and 5 plants were weighed per Petri dish and averaged.

### 2.2.2. Seedling Experiment

After 120 h of the first drought treatment, leaf age, plant height, 1st leaf from top leaf area, 2nd leaf from top leaf area, shoot dry weight, root length, root dry weight and root shoot ratio were measured for each variety. Using the repeated drought method, the drought survival proportion for each species was calculated based on the seedling survival rate after the first drought and the seedling survival rate after the second drought.

$$\frac{seedling\ survival\ rate\ after\ the\ first\ drought + seedling\ survival\ rate\ after\ the\ second\ drought}{2}$$

### 2.2.3. Adult Experiment

Specific leaf weight: 5 randomly selected representative points in the plots at the tillering, booting, heading and filling stages, 1 point at each point, decompose the leaves and measure their area, then kill at 105 °C for 30 min, dry at 80 °C to a constant weight and weigh, which was determined using the following Formula (1):

$$specific\ leaf\ weight = \frac{leaf\ dry\ weight}{leaf\ area}.$$

Photosynthetic indicators: Net photosynthetic rate, stomatal conductance, transpiration rate and intercellular $CO_2$ concentration were measured at the tillering, booting, heading and filling stages, using a portable photosynthesis meter (LI-6400 model, LI-COR, Lincoln, NE, USA), from 9:00 a.m. to 11:00 in the morning on a clear and windless day, with three replicates per treatment. The water use efficiency (WUE) was calculated by Pn/Tr. The mean values were calculated [18].

Measurement of proline content: Proline content was determined using the sulphosalicylic acid method [19].

Measurement of protective enzyme activities: The SOD activity was determined following the nitrogen blue tetrazolium method, POD activity following the guaiacol method, and the CAT activity following the ultraviolet absorption method [20]. Measurement of malondialdehyde content: The malondialdehyde content was determined by the thiobarbituric acid color development method [21].

### 2.2.4. Soluble Protein Content

Fifty milligrams of freeze-dried powdered material was soaked in distilled water, and the solution was stirred for 30 s, settled for 30 min, and centrifuged for 5 min at $4000 \times g$, and then 1 mL was transferred to a polypropylene tube. Coomassie brilliant blue G-250 was combined with 1 mL supernatant, and the absorbance was measured at 595 nm, within 20 min after the reaction [22].

### 2.2.5. Determination of Yield

The panicle in review number per plant, grain number per panicle, seed-setting rate, and kernel weight were measured to calculate yield.

### *2.3. Data Statistics and Analysis*

All data were analyzed using Microsoft Excel 2021 software (Redmond, WA, USA); Duncan's with SPSS 24 (Chicago, IL, USA) was used for significance of differences analysis; all plots were conducted using Sigma plot 14.0 (Palo Alto, CA, USA), SPSS and R. The germination period was evaluated using a composite evaluation of trait indicators using membership function, with each indicator calculated using the following formula.

$$M(X\_j) = (X\_j - X\_min)/(X\_max - X\_min) \quad j = 1,2,3, \ldots ,n \tag{1}$$

$\mu(X\_j)$ denotes the value of the membership function of the $j$th composite indicator, $X\_j$ denotes the value of the $j$th composite indicator, $X\_max$ denotes the maximum value of the $j$th composite indicator and $X\_min$ denotes the minimum value of the $j$th composite indicator.

$$W_j = r_j / \sum_{j=1}^{n} r_j \quad j = 1,2,3, \ldots ,n \tag{2}$$

$W_j$ denotes the weight of the $j$th composite indicator among all composite indicators, and $r_j$ is the coefficient of variation of the contribution of the $j$th composite indicator of the species.

$$D = \sum_{j=1}^{n} \left[ \mu(X_j) \times W_j \right] \quad j = 1,2,3, \ldots ,n \tag{3}$$

D indicates the combined assessment value of the drought resistance of each variety.

$$Y_{LR} = (Y_{ic} - Y_{id})/Y_{ic} \tag{4}$$

$i$ is the variety, $Y_{ic}$ is the yield of the control under water crop conditions for each variety, $Y_{id}$ is the yield under drought conditions for each variety, $Y_{LR}$ is rate of loss yield (%).

Hierarchical cluster analysis was conducted following the minimum variance method of Ward (1963), based on squared Euclidean distances [23].

## 3. Results

### *3.1. Evaluation of Drought Tolerance in 69 Rice during Germination Stage*

A total of 69 rice variety sources were evaluated; the relative values of shoot length, root length, root number, shoot weight, root weight, coleoptile length and germination rate were evaluated. These results suggested that rice varieties are differentially drought tolerant (Table 1). Additionally, the coefficient of variation for relative coleoptile length was the smallest at 28.60, accounting for the least weight, while the coefficients of variation for

relative root weight, relative root length and relative shoot weight were 60.67, 58.25 and 57.57, respectively.

**Table 1.** Overall evaluation value (D) and ranking of 69 rice varieties in germination trials in 2019.

| No. | Varieties | M (RCL) | μ (RSL) | M (RRL) | μ (RRN) | μ (RSW) | μ (RRW) | μ (RGR) | D Value | Rank |
|-----|-----------|---------|---------|---------|---------|---------|---------|---------|---------|------|
| 1 | Jipinlongxiang 180 | 0.6309 | 0.2002 | 0.1796 | 0.1823 | 0.0505 | 0.1285 | 0.2288 | 0.1960 | 63 |
| 2 | Daohuaxiang 8 | 0.1612 | 0.0925 | 0.0244 | 0.1787 | 0.0002 | 0.0454 | 0.1345 | 0.0822 | 68 |
| 3 | Longdao 16 | 0.0000 | 0.1829 | 0.0413 | 0.0858 | 0.1001 | 0.1433 | 0.2148 | 0.1149 | 67 |
| 4 | Longdao 20 | 0.9184 | 0.3172 | 0.1911 | 0.1155 | 0.4175 | 0.1633 | 0.2779 | 0.2999 | 55 |
| 5 | Songjing16 | 0.7334 | 0.1376 | 0.1220 | 0.1339 | 0.1229 | 0.1095 | 0.4790 | 0.2188 | 61 |
| 6 | Fangyuan 18 | 0.8970 | 0.4215 | 0.4081 | 0.8281 | 0.3750 | 0.2973 | 0.4290 | 0.4900 | 43 |
| 7 | Fangxiang 2 | 0.8247 | 0.8424 | 0.4620 | 0.4618 | 0.8565 | 0.7484 | 0.5476 | 0.6712 | 25 |
| 8 | Zhenzhuxiang | 0.3705 | 0.2538 | 0.3418 | 0.5356 | 0.3893 | 0.2746 | 0.4568 | 0.3706 | 49 |
| 9 | Daohuaxiang 7 | 0.9626 | 0.4581 | 0.3748 | 0.5262 | 0.6093 | 0.3647 | 0.9950 | 0.5723 | 39 |
| 10 | Shouzhe 918 | 0.7511 | 0.5939 | 0.5403 | 0.7783 | 0.8352 | 0.3053 | 0.5400 | 0.6080 | 38 |
| 11 | Wuyoudao 4 | 0.9187 | 0.4020 | 0.4078 | 0.3628 | 0.2920 | 0.2313 | 0.3383 | 0.3836 | 46 |
| 12 | Zhongke 804 | 0.8045 | 0.5512 | 0.1707 | 0.8536 | 0.4683 | 0.3367 | 0.9751 | 0.5588 | 40 |
| 13 | Longyang 16 | 0.4049 | 0.2277 | 0.2099 | 0.2630 | 0.2250 | 0.1750 | 0.1981 | 0.2310 | 59 |
| 14 | Longyang 20 | 0.9343 | 0.1888 | 0.0601 | 0.4712 | 0.0787 | 0.2982 | 0.1439 | 0.2644 | 57 |
| 15 | Longyang 21 | 0.6499 | 0.3911 | 0.2316 | 0.5906 | 0.3003 | 0.2353 | 0.5018 | 0.3885 | 44 |
| 16 | Longyang 06-6 | 0.3871 | 0.4652 | 0.2814 | 0.4934 | 0.3443 | 0.1737 | 0.5272 | 0.3722 | 48 |
| 17 | Longyang 13 | 0.9139 | 1.0000 | 0.4863 | 0.9658 | 0.8806 | 0.4600 | 0.8571 | 0.7752 | 12 |
| 18 | Suidao 9 | 0.1670 | 0.0000 | 0.0591 | 0.0000 | 0.0000 | 0.0494 | 0.0541 | 0.0383 | 69 |
| 19 | Shengyu 1 | 0.8409 | 0.2547 | 0.3763 | 0.4264 | 0.3726 | 0.2098 | 0.4838 | 0.3876 | 45 |
| 20 | Suidao 3 | 0.6841 | 0.1403 | 0.1544 | 0.1867 | 0.2015 | 0.2141 | 0.3345 | 0.2402 | 58 |
| 21 | Suijing 9 | 0.5204 | 0.1805 | 0.1685 | 0.2927 | 0.5193 | 0.1150 | 0.2810 | 0.2787 | 56 |
| 22 | Muyudao 49 | 0.9378 | 0.4240 | 0.4441 | 0.9125 | 0.4154 | 0.5442 | 0.8931 | 0.6199 | 33 |
| 23 | Yuxiang 3 | 0.8355 | 0.2896 | 0.1986 | 0.2844 | 0.2999 | 0.4070 | 0.5763 | 0.3760 | 47 |
| 24 | Lianyu 1013 | 0.4334 | 0.1463 | 0.0469 | 0.0854 | 0.1502 | 0.1136 | 0.5764 | 0.1942 | 64 |
| 25 | Lianyu 06124 | 0.7799 | 0.2814 | 0.5246 | 0.1725 | 0.2628 | 0.1509 | 0.4850 | 0.3457 | 53 |
| 26 | Suijing 28 | 0.8105 | 0.2148 | 0.2195 | 0.2050 | 0.3481 | 0.2295 | 0.7153 | 0.3499 | 52 |
| 27 | Suiyu 117463 | 0.8556 | 0.8658 | 0.3969 | 0.7156 | 0.7463 | 0.8760 | 0.9587 | 0.7607 | 13 |
| 28 | Longjing 1656 | 0.8817 | 0.2801 | 0.1711 | 0.5272 | 0.2375 | 0.2006 | 0.6157 | 0.3697 | 50 |
| 29 | Longjing 1525 | 0.5950 | 0.7424 | 0.0000 | 0.1295 | 0.0601 | 0.0000 | 0.0000 | 0.1878 | 65 |
| 30 | Zhongkefa 6 | 0.9417 | 0.3619 | 0.2610 | 0.1570 | 0.2673 | 0.1746 | 0.7163 | 0.3599 | 51 |
| 31 | Beidao 1 | 0.8220 | 0.2894 | 0.1159 | 0.1445 | 0.1464 | 0.0819 | 0.0719 | 0.1962 | 62 |
| 32 | Longnian 588 | 0.8011 | 0.1849 | 0.3211 | 0.0379 | 0.1856 | 0.1807 | 0.1492 | 0.2294 | 60 |
| 33 | Qingling 998 | 0.6320 | 0.2281 | 0.1595 | 0.2245 | 0.8963 | 1.0000 | 0.2766 | 0.4936 | 42 |
| 34 | Heizhenzhu | 0.9122 | 0.6937 | 0.2912 | 0.6694 | 0.9089 | 0.3779 | 0.6560 | 0.6182 | 35 |
| 35 | Jiyang 108 | 0.6662 | 0.9885 | 0.5516 | 0.7633 | 0.6106 | 0.3979 | 0.9257 | 0.6890 | 20 |
| 36 | Baijing 1 | 0.4751 | 0.9774 | 0.2953 | 0.8139 | 0.5761 | 0.6094 | 0.9421 | 0.6702 | 27 |
| 37 | Tongke 37 | 0.7401 | 0.8379 | 0.3451 | 0.6576 | 0.6889 | 0.6462 | 0.9852 | 0.6854 | 23 |
| 38 | Jinian 6 | 0.7858 | 0.8829 | 0.3233 | 0.8260 | 0.9593 | 0.2185 | 0.9796 | 0.6874 | 22 |
| 39 | Jinongda 828 | 0.7915 | 0.6463 | 0.4796 | 0.6508 | 0.7509 | 0.3111 | 0.8645 | 0.6193 | 34 |
| 40 | Lvdao 177 | 0.7386 | 0.8022 | 0.4218 | 0.9599 | 0.9219 | 0.5740 | 0.9467 | 0.7575 | 14 |
| 41 | Jinongda 603 | 0.9640 | 0.5951 | 0.4496 | 0.6474 | 0.6447 | 0.3333 | 0.9407 | 0.6165 | 36 |
| 42 | Jingu 119 | 0.9449 | 0.9138 | 0.5347 | 0.7005 | 0.9411 | 0.8425 | 0.9031 | 0.8138 | 8 |
| 43 | Changjing 616 | 1.0000 | 0.8851 | 0.4980 | 0.8325 | 0.8776 | 0.6608 | 0.9740 | 0.7955 | 10 |
| 44 | Qinglin 168 | 0.8576 | 0.6253 | 0.5646 | 0.5198 | 0.8284 | 0.3830 | 0.9347 | 0.6496 | 28 |
| 45 | Jinongda 899 | 0.7638 | 0.6087 | 0.3502 | 0.5350 | 0.9126 | 0.4883 | 0.9583 | 0.6414 | 30 |
| 46 | Jinongda 858 | 0.7638 | 0.6087 | 0.3502 | 0.5350 | 0.9126 | 0.4883 | 0.9583 | 0.6414 | 30 |
| 47 | Wokeshou 1 | 0.6665 | 0.8664 | 1.0000 | 0.7333 | 0.9146 | 0.6344 | 0.9700 | 0.8341 | 6 |
| 48 | Songjing 22 | 0.7449 | 0.9537 | 0.4968 | 0.6314 | 0.9213 | 0.6840 | 0.8100 | 0.7454 | 17 |
| 49 | Longyang19 | 0.9042 | 0.9270 | 0.7829 | 0.8662 | 0.9353 | 0.6762 | 0.8477 | 0.8413 | 5 |
| 50 | Yilongdundao | 0.8112 | 0.5459 | 0.4338 | 0.6693 | 0.6619 | 0.6471 | 0.6275 | 0.6135 | 37 |

| No. | Varieties | M (RCL) | μ (RSL) | M (RRL) | μ (RRN) | μ (RSW) | μ (RRW) | μ (RGR) | D Value | Rank |
|-----|-----------|---------|---------|---------|---------|---------|---------|---------|---------|------|
| 51 | Longdao 202 | 0.7671 | 0.6613 | 0.8701 | 1.0000 | 0.9504 | 0.7785 | 0.9897 | 0.8625 | 3 |
| 52 | Suijing 27 | 0.9427 | 0.7212 | 0.7113 | 0.9013 | 0.8388 | 0.7641 | 0.8677 | 0.8090 | 9 |
| 53 | Suijing 18 | 0.9731 | 0.8774 | 0.5578 | 0.8636 | 0.9666 | 0.8139 | 0.9629 | 0.8455 | 4 |
| 54 | Longjing 21 | 0.9537 | 0.7489 | 0.6008 | 0.6715 | 0.9172 | 0.5740 | 0.8951 | 0.7458 | 16 |
| 55 | Longdao 18 | 0.9815 | 0.9916 | 0.7993 | 0.9530 | 0.9973 | 0.6781 | 0.9508 | 0.8962 | 2 |
| 56 | Hongke 67 | 0.8751 | 0.4835 | 0.6246 | 0.6373 | 0.6911 | 0.8671 | 0.7593 | 0.6945 | 19 |
| 57 | Hongke 88 | 0.9230 | 0.3144 | 0.1681 | 0.1598 | 0.1931 | 0.2128 | 0.5000 | 0.3034 | 54 |
| 58 | Hongke 8 | 0.7797 | 0.7218 | 0.4722 | 0.9712 | 0.9426 | 0.4298 | 0.9554 | 0.7386 | 18 |
| 59 | Jinongda 138 | 0.9027 | 0.9554 | 0.7192 | 0.7314 | 0.8935 | 0.7048 | 0.3453 | 0.7488 | 15 |
| 60 | Jinongda 168 | 0.8510 | 0.6178 | 0.5179 | 0.4992 | 0.8165 | 0.4396 | 0.9168 | 0.6427 | 29 |
| 61 | Jinongda 738 | 0.5919 | 0.8530 | 0.5994 | 0.7565 | 0.8343 | 0.2564 | 0.9832 | 0.6889 | 21 |
| 62 | Tongyuanxiang 518 | 0.9509 | 0.0328 | 0.0277 | 0.0813 | 0.0789 | 0.1257 | 0.4172 | 0.1850 | 66 |
| 63 | Jinongda 838 | 0.2389 | 0.9577 | 0.5148 | 0.6460 | 0.9296 | 0.3496 | 0.9290 | 0.6704 | 26 |
| 64 | Hongke 57 | 0.8863 | 0.9008 | 0.3892 | 0.8073 | 0.1008 | 0.6064 | 1.0000 | 0.6363 | 32 |
| 65 | Tonghuayuan | 0.9866 | 0.8607 | 0.5160 | 0.8125 | 0.9305 | 0.7871 | 0.9947 | 0.8234 | 7 |
| 66 | Tieganxiang 2 | 0.8261 | 0.7729 | 0.5991 | 0.8294 | 1.0000 | 0.5521 | 0.9386 | 0.7775 | 11 |
| 67 | Hanxiang7 | 0.9127 | 0.9378 | 0.8981 | 0.9245 | 0.9858 | 0.7823 | 0.9852 | 0.9148 | 1 |
| 68 | Longqingdao21 | 0.8369 | 0.5326 | 0.6387 | 0.5620 | 0.6749 | 0.6960 | 0.9633 | 0.6844 | 24 |
| 69 | Miaodao 74 | 0.8132 | 0.3719 | 0.2691 | 0.5990 | 0.4483 | 0.3745 | 0.9856 | 0.5155 | 41 |
| Coefficient of variation | | 28.60 | 53.62 | 58.25 | 52.95 | 57.57 | 60.67 | 45.80 | | |
| Weight | | 0.0800 | 0.1500 | 0.1629 | 0.1481 | 0.1610 | 0.1697 | 0.1281 | | |

Note. RCL, relative coleoptile length; RSL, relative shoot length; RRL, relative root length; RRN, relative root number; RSW, relative shoot weight; RRW, relative root weight; RGR, relative germination potential.

The 69 rice accessions were clustered into three groups, based on the cluster analysis: strongly tolerant, tolerant and strongly sensitive (Figure 1). The first group contained ten strongly tolerant accessions (No. 55, 67, 51, 49, 53, 47, 65, 43, 42, 52), the second group contained 33 tolerant accessions (No. 9, 12, 69, 6, 33, 58, 59, 48, 54, 17, 66, 27, 40, 7, 36, 63, 56, 37, 68, 38, 35, 61, 44, 64, 60, 45, 46, 10, 41, 50, 34, 22, 39), the third contained 26 strongly sensitive accessions No. 31, 1, 24, 62, 29, 32, 13, 20, 5, 21, 14, 57, 4, 28, 8, 16, 23, 19, 15, 11, 26, 25, 30, 3, 2, 18 (Figure 1; Table 1).

### 3.2. Effect of Drought Stress on Various Agronomic Traits at Seedling Stage

Twenty-five varieties were screened from the germination stage for seedling drought resistance screening. The 25 seedling varieties were clustered by drought survival proportion (Figure 2, Table S1) and could be divided into three broad classes at Euclidean clustering distances equal to 5.Category I is a strongly drought-resistant class, with 11 varieties, including No. 49, 52, 65, 67, 1, 43, 55, 53, 42, 54, 51. Category II is for drought-resistant varieties, with five varieties, including No. 3, 29, 20, 60, 47. Category III is the non-drought tolerant varieties, with nine in total including No. 32, 2, 12, 62, 31, 18, 5, 24, 57.

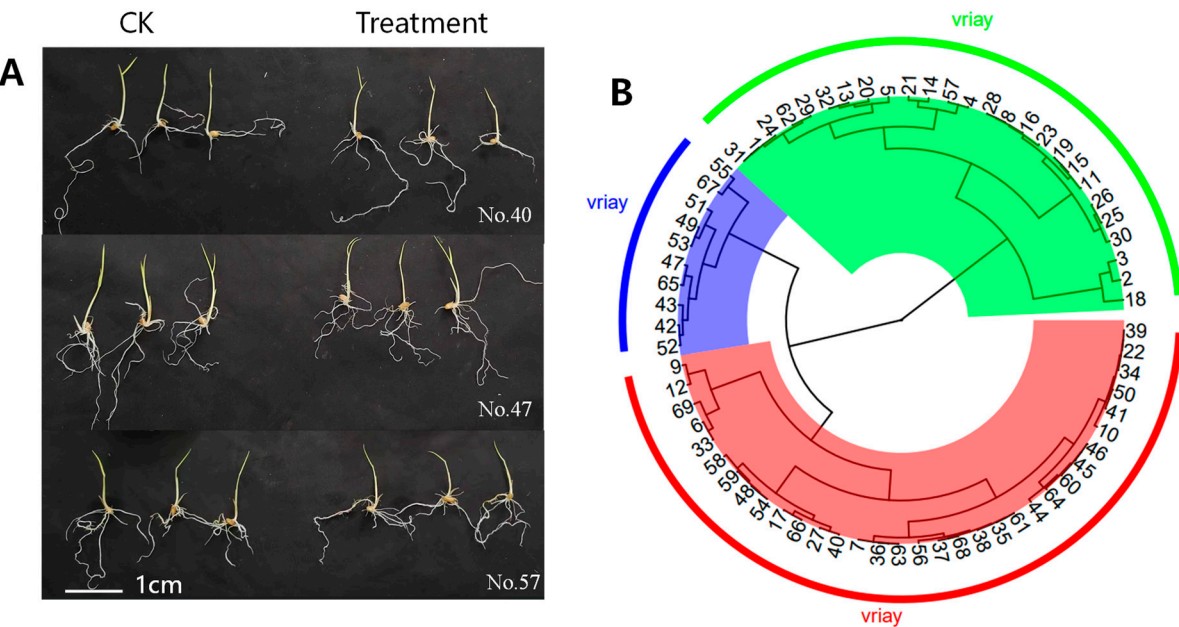

**Figure 1.** The growth of rice treated with 20% concentration of PEG-6000 solution (**A**) and cluster analysis of 69 varieties at germination stage in 2019 (**B**). The colors represent the classification of the different drought-resistant varieties, Green for strongly sensitive varieties. Red for tolerant varieties. Blue for strongly tolerant varieties.

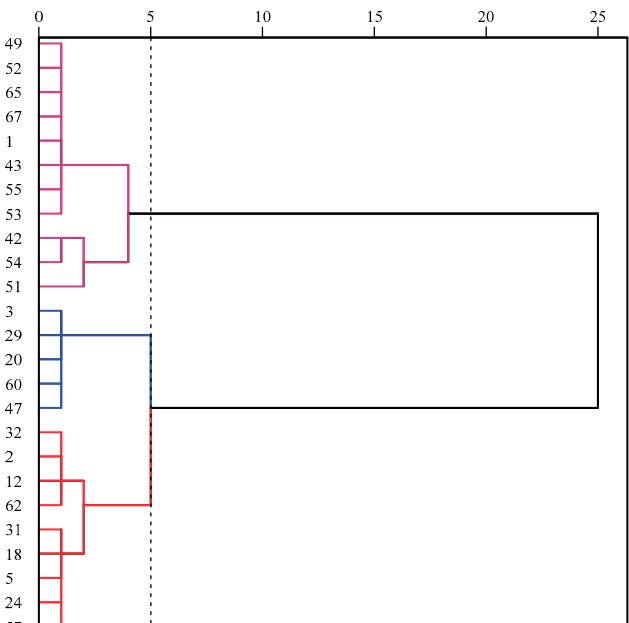

**Figure 2.** Cluster analysis of 25 varieties at seedling stage in 2019. At a euclidean distances of 5, the 25 varieties are divided into 3 categories, with one color representing one category.

Drought survival proportion is an important identification method for evaluating seedling and field [24–26]. There is a correlation between each indicator and repeated survival at the seedling stage (Table S2). The mean values of plant height, 1st leaf from top leaf length and root length decreased significantly under dry cultivation conditions by 11.52%, 13.59% and 11.31%, respectively, while the mean values of above-ground dry weight remained unchanged, and the root shoot ratio increased (Table S3). Under control conditions, the coefficients of variation for 1st leaf from top leaf area, root dry weight and root shoot ratio were the highest under control conditions, at 39.67, 35.07 and 39.85,

respectively. The highest coefficients of variation were found for 2nd leaf from top leaf area, root dry weight and root shoot ratio at 30, 30.19 and 38.01, respectively. Drought has a greater effect on plant height, 2nd leaf from top leaf width, 2nd leaf from top leaf area and leaf age.

### 3.3. Effects of Drought on Antioxidant Activity of Rice under Dry Cultivation

Based on the field performance of different varieties, 'Suijing 18', 'Changjing 616', 'Hongke 88' and 'Tongyuanxiang 518' were selected to clarify the physiological characteristics of the drought resistance of the varieties. Several antioxidant parameters were investigated to assess the impact of dry cultivation on the rice varieties (Figure 3). The SOD activity of the strong drought tolerant varieties increased as the period progressed. The strongly drought tolerant varieties were significantly higher than the non-drought tolerant varieties at the booting, heading, filling stages and reached a significant difference, while the non-drought tolerant variety 'Hongke 88' was significantly higher than the other varieties at the tillering stage (Figure 3A). The SOD activity of 'Changjing 616' was higher than that of 'Suijing 18' in all periods, with the exception of the heading stage. The POD activity of different varieties under drought stress reached the maximum at the filling stage, and 'Changjing 616' was significantly higher than other varieties (Figure 3B). At the heading stage 'Suijing 18' was significantly higher than the other treatments. No significant difference was found between the varieties at the booting stage, and at the tillering stage Tongyuan Xiang 518 was significantly higher than the other varieties. The CAT activity of different varieties under drought stress showed a trend of increasing and then decreasing, reaching a maximum value at the booting stage, and the CAT activity of strongly drought-resistant varieties was higher than that of non-drought-resistant varieties during the whole growing period, reaching a significant difference at the tillering and booting stages (Figure 3C). These results indicate that strongly drought tolerant varieties were actively resistant to oxidative damage.

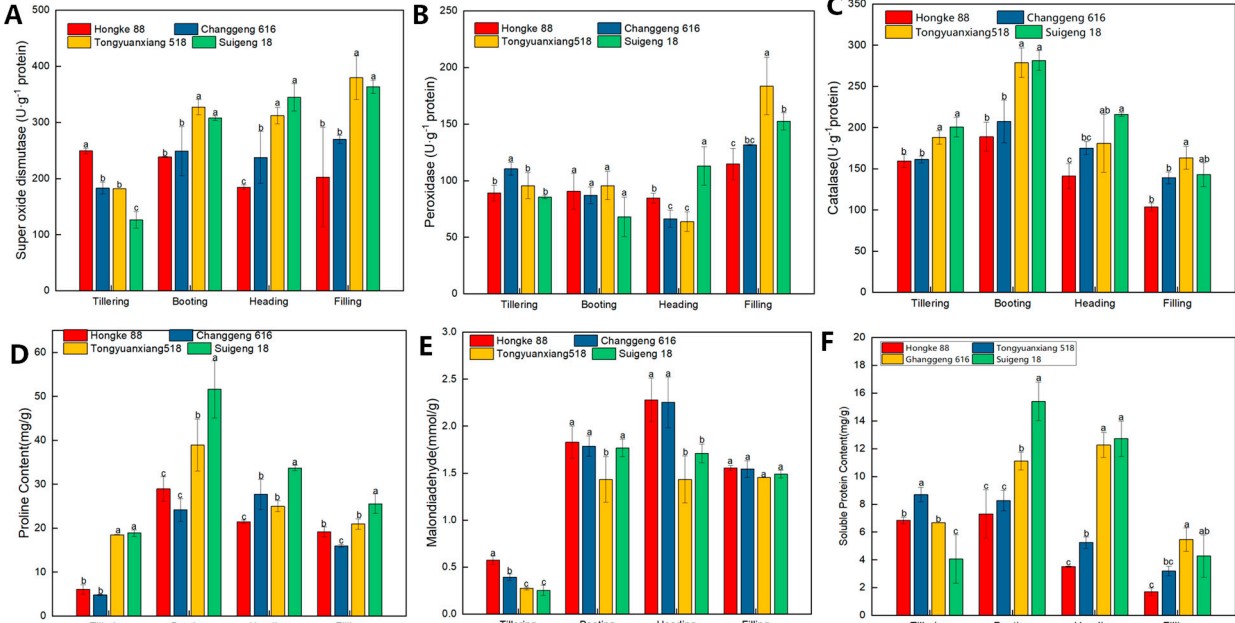

**Figure 3.** Changes in the physiological characteristics of different varieties of rice after under dry cultivation. SOD (superoxidase dismutas) (**A**), POD (Peroxidase) (**B**), CAT (catalyses) (**C**) activities. Proline (**D**), MDA (Malondialdehyde) (**E**) and soluble protein (**F**), Vertical bars represent the S.E. of the means (n = 3). Within each stage of an experiment, means bars marked by the same letters are not significantly different at $p < 0.05$.

Drought stress caused significant effects on proline (Pro) and malondialdehyde (MDA) in different drought-tolerant varieties. The Pro content of different varieties showed a trend of first increasing and then decreasing, with all varieties reaching a maximum at the booting stage, with the exception of the 'Tongyuanxiang 518' variety. The Pro content of the strongly drought-tolerant varieties was significantly higher than that of the non-drought-tolerant varieties at the tillering and booting stages (Figure 3D). MDA is the end product of the degree of peroxidation of cellular membrane lipids and reflects the degree of damage to crop cells and the resistance of the variety. The MDA content of the non-drought tolerant type was higher than that of the strongly drought tolerant variety; this was particularly significant at the tillering and heading stages. There was no significant difference between 'Suijing 18' and 'Changjing 616', with the exception of the booting stage. The non-drought tolerant variety, 'Hongke 88', was higher than 'Tongyuanxiang 518' during the whole growing stage (Figure 3E). We have observed that with the increase in time, the soluble protein content initially increased, but later decreased in growth (Figure 3F). The soluble protein content of the non-drought tolerant type was lower than that of the strongly drought tolerant variety, which was particularly significant at the booting and heading stages.

### 3.4. Effects of Drought on Photosynthetic Parameters of Rice under Dry Cultivation

The net photosynthetic rate showed an overall trend of increasing and then decreasing. The strongly drought-resistant variety, Changjing 616, maintained a high level of net photosynthetic rate throughout the whole stage. At the tillering stage, the net photosynthetic rate of the strongly drought-resistant variety, 'Suijing 18', was lower than that of the non-drought-resistant variety, and the difference was not significant at the booting stage. As leaf senescence progressed, the net photosynthetic rate of the strongly drought-tolerant varieties was higher than that of the non-drought-tolerant varieties at the heading and filling stages, reaching a significant difference at the filling stage (Figure 4A). There was no significant difference in the intercellular $CO_2$ concentration at the tillering and heading stage, and the non-drought tolerant type was higher than the strongly drought tolerant variety at the booting and filling stages (Figure 4B). The magnitude of stomatal conductance influences the transpiration rate. The trends of the stomatal conductance and transpiration rate were consistent with photosynthesis (Figure 4C,D). Water-use efficiency was significantly higher in 'Hongke 88' than the other varieties at the tillering stage, highest in 'Tongyuanxiang 518' at the booting stage, and higher in non-drought tolerant varieties than strongly drought tolerant varieties at the heading and filling stages, reaching significant differences at the filling stage (Figure 4E). The specific leaf weight of the four varieties did not change significantly in each period—'Changjing 616' > 'Suijing 18' > 'Tongyuanxiang 518' > 'Hongke 88'—and the specific leaf weight of the strongly drought-resistant varieties was significantly higher than that of the non-drought-resistant varieties in the whole growing period (Figure 4F), indicating that the leaf weight per unit area of the strongly drought-resistant varieties was higher than that of the non-drought-resistant varieties.

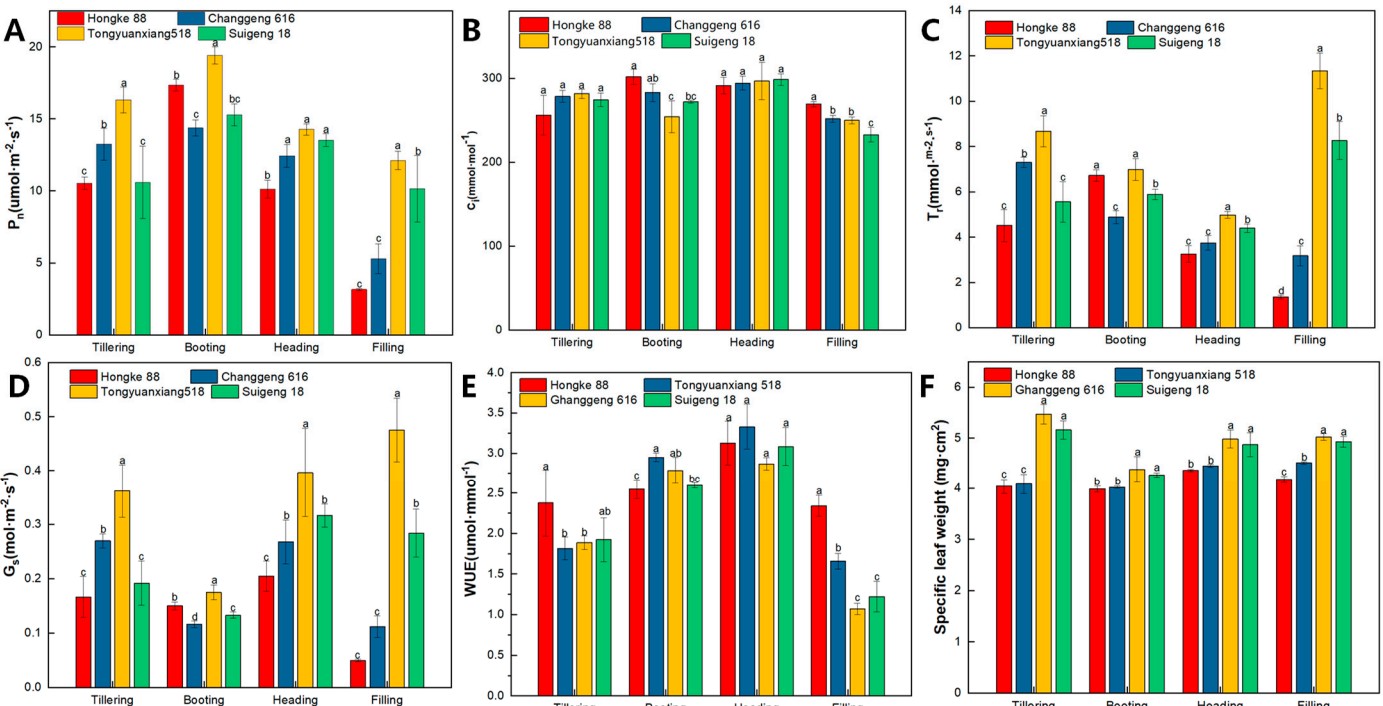

**Figure 4.** Changes in photosynthetic physiological indicators of different varieties under drought stress in 2021. (**A**), Net photosynthetic rate; (**B**), Intercellular $CO_2$ concentration; (**C**), Transpiration rate; (**D**), Stomatal conductivity; (**E**), Water-use efficiency; (**F**), Specific leaf weight; Vertical bars represent the S.E. of the means (n = 3). Within each stage of an experiment, means bars marked by the same letters are not significantly different at $p < 0.05$.

*3.5. Differences in Yield and Yield Components of Different Drought-Resistant Varieties*

Under water and dry cultivation conditions, rice varieties with different characteristics differ in yield (Table 2). 'Changjing 616' has the highest yield under dry condition, primarily through the number of grains on the spike. The more drought-resistant the variety, the lower the yield loss rate. Strongly drought-resistant varieties were lower drought-resistant than non-drought-resistant varieties, and the yield loss rates were reduced by 42.71–66.26%. Under drought conditions, the spike numbers, grains numbers, seed setting percentage, 1000-grain weight and yield all decreased, and the degree of decrease varied, with the spikes number decreasing from 4.76% to 23.81%; grains numbers from 25.05% to 51.58%; seed setting percentage from 4.36% to 8.20%; and 1000-grain weight from 4.86% to 22.90%. The spike numbers were the main factor in the yield decline, while the seed setting percentage had the least effect. The strongly drought-resistant varieties, 'Changjing 616' and 'Suijing 18', were identified according to the yield loss rate, and the non-drought-resistant varieties 'Hongke 88' and 'Tongyuanxiang 518' were also identified.

**Table 2.** Yield and yield components of different rice varieties.

| Treatment | | Varieties | Panicles/m$^2$ | Per Panicle | Seed-Setting Rate (%) | 1000–Grain Weight/g | Yield /(kg·ha) | Yield Loss (%) |
|---|---|---|---|---|---|---|---|---|
| Control | Non-drought tolerant varieties | Hongke 88 | 380.00 ± 11.25 ab | 107.39 ± 2.30 b | 97.01 ± 0.98 a | 25.20 ± 0.94 a | 9087.21 ± 306.90 a | 66.26 |
| | | Tongyuanxiang 518 | 344.00 ± 18.27 bc | 97.83 ± 2.83 b | 96.03 ± 0.78 abc | 27.14 ± 0.34 a | 8470.60 ± 139.14 a | 62.82 |
| | Drought-resistant varieties | Jinongda 168 | 302.40 ± 20.16 c | 144.61 ± 15.69 a | 94.75 ± 0.88 cd | 22.84 ± 0.92 b | 8813.32 ± 512.35 a | 59.91 |
| | | Suidao 3 | 380.48 ± 11.36 ab | 96.83 ± 3.62 b | 93.98 ± 0.34 d | 26.39 ± 0.72 a | 8857.89 ± 94.90 a | 59.36 |
| | Strongly drought-resistant varieties | Changjing 616 | 308.56 ± 7.90 c | 146.55 ± 2.04 a | 95.21 ± 0.34 bcd | 21.39 ± 0.20 c | 8450.70 ± 406.11 a | 46.56 |
| | | Suijing 18 | 397.33 ± 55.06 a | 77.72 ± 10.84 c | 96.48 ± 1.08 ab | 26.18 ± 2.56 a | 6557.83 ± 386.76 b | 42.71 |
| Drought | Non-drought tolerant varieties | Hongke 88 | 341.33 ± 18.48 a | 52.00 ± 1.00 d | 89.06 ± 2.17 a | 19.43 ± 0.57 c | 3066.36 ± 416.29 c | —— |
| | | Tongyuanxiang 518 | 256.00 ± 16.00 c | 58.72 ± 0.25 c | 91.84 ± 1.24 a | 25.82 ± 1.64 a | 3149.69 ± 174.98 c | —— |
| | Drought-resistant varieties | Jinongda 168 | 264.80 ± 8.80 bc | 81.60 ± 5.53 b | 90.31 ± 2.13 a | 19.27 ± 1.99 c | 3532.98 ± 305.47 bc | —— |
| | | Suidao 3 | 360.00 ± 14.40 a | 52.50 ± 2.50 d | 88.72 ± 3.10 a | 22.49 ± 0.08 b | 3599.64 ± 346.38 bc | —— |
| | Strongly drought-resistant varieties | Changjing 616 | 293.87 ± 8.78 b | 90.17 ± 1.44 a | 89.60 ± 2.22 a | 19.52 ± 0.31 c | 4516.22 ± 375.24 a | —— |
| | | Suijing 18 | 346.67 ± 9.24 a | 58.25 ± 0.75 c | 91.03 ± 2.46 a | 20.53 ± 0.17 c | 3757.20 ± 209.92 b | —— |

Note: Different lowercase letters indicate significant differences at *p* < 0.05.

## 4. Discussion

### 4.1. Whole Growing Period Screens for Drought Resistance Is a Necessary Process for Classifying Varieties for Resistance

Most of the previous evaluations of germplasm resources were carried out at the germination and seedling stages [27–29], with the germination stage being identified and screened by polyethylene glycol (PEG-6000), simulating drought in a simpler, for a more controlled and shorter period [30]. The membership function is an important method widely used to evaluate crop stress tolerance levels [31]. Therefore, in this research, 69 varieties were evaluated for drought resistance under simulated drought conditions in PEG-6000 solution at the germination stage; 10 strongly drought-resistant, 5 drought-resistant and 10 non-drought-resistant varieties were selected using the membership function method. Seedling drought resistance identification has the advantages of easy in vivo experimenting, low environmental impact and a short cycle time [11]. Repeated drought in seedlings is a common method, and the level of repeated survival can reflect the degree of drought resistance in seedlings of the crop [30]. The above 25 varieties were used for cluster analysis at the seedling stage based on drought survival proportion, and 11 strongly drought-resistant, 5 drought-resistant and 9 non-drought -resistant varieties were selected. 9 strongly drought-resistant varieties, 2 drought-resistant and 7 non-drought-resistant varieties were screened together at germination and seedling stages. The varieties screened for drought resistance at germination and seedling stages were not completely consistent, which indicates that the drought resistance screening of varieties from germination or seedling stages alone is not sufficient; in other words, drought resistance at the germination and seedling stages does not fully reflect the drought resistance level of varieties. The identification of drought resistance at the adult stage is usually based on yield indicators, such as drought tolerance index, sensitivity index and yield loss rate [31], which is a method that can effectively respond to the strength of drought resistance in rice. To ensure the accuracy of the results, drought-tolerant rice varieties obtained at the germination and seedling stages still need to be grown in the field [32]. Two strongly drought-resistant, drought-resistant and non-drought-resistant varieties were selected at the germination and seedling stages, and the strongly drought-resistant varieties 'Changjing 616' and 'Suijing 18' were finally determined according to the yield loss rate at the adult stage. The grain yield of the group of strongly drought-resistant varieties is higher, with 'Changjing 616' increasing primarily through the number of spikes and 'Suijing 18' primarily through the number of tillers, with yields of 4516.22 kg/ha and 3757.20 kg/ha, respectively, under drought conditions. The drought resistance of the varieties screened at the germination and seedling stages was consistent with the performance at the adult stage. This suggests that screening at the germination and seedling stages, and the identification of drought resistance at the adult stage, are necessary for the classification of varieties for drought resistance.

### 4.2. Varietal Differences in Drought Resistance Are a Combination of Physiological and Metabolic Processes

The strength of drought resistance in rice is a quantitative trait; the mechanisms of drought resistance are extremely complex, and the expression of drought resistance is also complex and variable [33]. Seed germination and growth periods are the most critical elements of their individual growth and development, and the growth of shoots and roots and their biomass are significantly affected by drought stress during this period [34]. This view was also demonstrated in this experiment. The crop root system is the main organ for water and nutrient uptake and transport, and morphological changes in the root system significantly affect above-ground growth, development and yield [35,36]. Under soil drought stress conditions, crops increase their root growth by improving root assimilate allocation in order to enhance water uptake, thereby increasing their root dry weight and root/shoot ratio [37]. Root length, root weight and root characteristics at the seedling stage of maize were highly significantly related to their drought resistance [38]. Wheat seedlings subjected to drought stress showed greater variation and higher heritability in root length,

root/shoot ratio and membrane permeability, also reflecting the drought resistance of the crop [36]. In this research, drought screening at the seedling stage was described primarily by growth and developmental and morphological indicators, and the drought survival proportion was highly significantly and positively correlated with root length and root dry weight, with results consistent with previous studies [36].

Plants have gradually evolved their own defense mechanisms in response to drought stress, from cells to ecosystems, through a variety of pathways, of which drought avoidance is one of the adaptive mechanisms for drought resistance in crops [1]. In this research, drought stress was found to accelerate the senescence process of leaves of non-drought tolerant varieties [39], completing filling to maturity approximately 7–10 days earlier. Drought also affects plant morphological indicators and a range of physiological and biochemical processes. Drought-tolerant varieties have high levels of both antioxidant enzyme activity and net photosynthetic rate, which can delay the damage caused by drought. In an adverse environment, the three enzymes POD, SOD and CAT work together to coordinate the maintenance of free radicals in the plant [40] and ensure the normal functioning of physiological and biochemical reactions in the plant. During the filling stage, the POD activity was higher in strongly drought-resistant varieties than in non-drought-resistant varieties, and SOD (except at the tiller stage) and CAT activities were higher in strongly drought-resistant varieties than in non-drought-resistant varieties. However, under severe drought stress conditions, the antioxidant enzyme system cannot scavenge the reactive oxygen species in time and cause cell membrane damage, and the MDA content can reflect the degree of lipid peroxidation and membrane injury and the MDA content usually increases under stress damage [24]. In general, stress-sensitive plants have a higher MDA content in response to drought stress than stress-resistant plants. In the present study, the MDA content of the strongly drought tolerant varieties were significantly lower than that of the non-drought tolerant varieties. Proline can accumulate to high concentrations without damaging cellular macromolecules. Therefore, it acts as a compatible osmolyte. Importantly, proline provides protection against membrane damage and protein denaturation during severe drought stress [41]. Proline accumulation is a physiological response of plants subjected to drought stress [42,43]. Our results regarding proline accumulation were in agreement with the results of studies cited above. In the present study, the proline content of the leaves was significantly increased, especially in strongly drought-resistant varieties. In conclusion, drought-intolerant varieties resist drought stress, primarily through drought avoidance, but this approach is unsuitable for promotion as a cultivar due to the shortening of the reproductive period, resulting in reduced yields. Strongly drought resistant varieties adapt to drought cultivation in rice, mainly by enhancing the resistance of the organism, and yield losses are small. The non-drought resistant rice varieties 'Hongke 88' and 'Tongyuanxiang 518' have yield losses of 66.26% and 62.82%, while the strongly drought resistant rice varieties Chang Japonica 616 and Sui Japonica 18 are both approximately 40%.

Previous research has shown that stomatal closure and Mesophyll conductance are the most significant causes of the inhibition of photosynthetic rates [44]. Higher apparent mesophyll conductance facilitates $CO_2$ transport and lower non-stomatal limitation [45]. The results of this experiment showed that the net photosynthetic rate of the strongly drought tolerant varieties could be maintained at a high level in the late reproductive period, with stomatal conductance and mesophyll conductance showing the same trend as the net photosynthetic rate, and water use efficiency showing the opposite trend to the net photosynthetic rate. This is because maintaining a high net photosynthetic rate is accompanied by a high transpiration rate [46] and a decrease in water use efficiency. Studies have shown that narrower leaves of strongly drought tolerant varieties can reduce their transpiration area and increasing leaf thickness can enhance their water storage capacity. The leaf weight per unit area of strongly drought tolerant varieties in this study is higher than that of non-drought tolerant varieties, indicating their higher water storage capacity and enhanced transpiration rate per unit area, which is consistent with the previous results.

The light response curve can effectively describe the change in leaf net photosynthetic rate at 0–2000 PAR [47], and the light saturation point of the strongly drought resistant variety is greater than that of the non-drought resistant variety, which indicates that the strongly drought resistant variety has a more efficient use of strong light and a higher change in net photosynthetic rate, which is consistent with the results above. The increase in photosynthesis in the later growth period of the process in the strongly drought resistant varieties compared to the non-drought resistant varieties may be related to the increased activity of antioxidant enzymes (SOD, POD, CAT) in the later growth period, which slows down the rate of leaf senescence due to reactive oxygen species damage and delays the later stages of photosynthesis. The higher specific leaf weight helps to increase leaf water content, maintain stomatal conductance and photosynthetic rate, increase dry matter transfer to the seeds and increase population seed weight.

## 5. Conclusions

In this experiment, 69 rice varieties in northeast China were screened for drought resistance at the germination, seedling and adult stages, and two strongly drought-resistant varieties, 'Changjing 616' and 'Suijing 18', were obtained by membership functions, correlation analysis and cluster analysis. Strongly drought tolerant rice varieties are adapted to rice under dry cultivation, primarily by enhancing the resistance of the organism, and have a higher specific leaf weight than non-drought tolerant varieties, which facilitates the maintenance of a higher photosynthetic rate and antioxidant enzyme activity under drought conditions. It can effectively delay leaf senescence caused by drought stress later in the growth period, prolong photosynthesis time, provide sufficient source reservoir accumulation for kernel filling and low yield loss, and is suitable as a popular variety of rice under dry cultivation. Therefore, understanding the distinct drought stress responses of different rice accessions may help breed crops adapted to the increasing drought stress and water saving agriculture. Such advances may maintain the yield of important food crops in the future.

**Supplementary Materials:** The following supporting information can be downloaded at: https://www.mdpi.com/article/10.3390/agronomy12112849/s1. Table S1: Number and name of the 25 varieties in the seedling test. Table S2: Correlation analysis of drought survival proportion and other traits in various rice varieties under drought stress. Table S3: Descriptive statistical analysis of drought stress on various traits in rice.

**Author Contributions:** M.Y. and Z.W. designed and supervised the project. X.W., B.C. and K.Y. analyzed the data and wrote the manuscript. W.L., P.T. and X.H. participated in the determination of physiological indexes. X.W., G.W. and Y.D. participated in the material preparation. All authors discussed the results and commented on the manuscript. All authors have read and agreed to the published version of the manuscript.

**Funding:** This study was supported by the following funding sources: Jilin Science and technology development plan project (20210101045JC, 20210509013RQ, 19190301061NY, 20200403016SF).

**Institutional Review Board Statement:** Not applicable.

**Informed Consent Statement:** Not applicable.

**Data Availability Statement:** The data presented in this study are available on request from the corresponding author. The data are not publicly available due to the involvement of other unpublished papers.

**Conflicts of Interest:** The authors declare no conflict of interest. The funders had no role in the design of the study; in the collection, analyses, or interpretation of data; in the writing of the manuscript, or in the decision to publish the result.

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
