# Peer review of "Physiological Characterization of Drought Responses and Screening of Rice Varieties under Dry Cultivation"

_agronomy, doi:10.3390/agronomy12112849_

Round 1

Reviewer 1 Report

-        Line 89: PEG is polyethylene glycol, please write it here, not mention it first in line 310.

-        Some mistypings occur, spaces are needed between numbers and dimensions between words, and after sentence ending dots. (line 93-95, 98, 100, 102, 103, 110-116, 123, 132, 137,

-        In lines 127-129, and 133 equations would look better in form of Equation 1, Equation 2,

-        The already named equations should be formatted to look more uniformed

-        Line 151-152, 162-163 Eq.(2) and Eq.(4) nomenclature should be revised and corrected e.g. using both forms Μ(X_j ) and μ(Xj) for the same factor is confusing for the readers.

-        Line 137 the sentence should be rephrased since words seem to be missing from it, and it makes harder to understand it. Maybe, it can be divided in 2 separete sentences.

-        Line 142 year is not needed in this form for the Xu et al. [18] reference.

-        Line 151-153: X_man should be X_max

-        Please, clarify CK. Is this the abbreviation for control? If yes, clarify it in the text (line 163).

-        Line 168-170 is recommended to replace into Methodology

-        In the Methodology section (probably after line 165), unfortunately, there is no part which presents the applied statistical analysis e.g. Cluster analysis mentioned in Results section (lines 183-184)

-        strange comma character is used in lines 185-190, and lines 199-202, line 224 these should be corrected.

-        For Fig. 1  part A measure scale bar would be needed.

-        Fig. 1 part B does not express the 3 clusters, but 2, I suggest to redo it in order to have a nice visual support.

-        Table 1 Field names should be edited before publish

-        Please correct 1 st to 1st, and 2 st to 2nd if you mean it this way.

-        Fig. 3 and 4: diagrams should be edited since it is hard to read the axe titles in this form, and Legend is not formatted the same, please redo those, without spacing errors.

-        Please, revise figure title text of Fig. 3.

-        Line 263: please correct „Chang- jing 616” to „ ’Changjing 616’ ”.

-        Table 2 is not well edited and hence it is hard to use.

-        Line 303: please, revise this Table title sentence.

Author Response

Point-by-point response to reviewers

Manuscript Title: Physiological characterization of drought responses and Screening of rice varieties under dry cultivation

Dear Editor:

First of all, we deeply appreciate the editor for your time and efforts. Based on your comments, we have uploaded the data in the manuscript and corrected the relevant experimental details. The revised parts are marked in different colors in the manuscript. Our specific response is as follows.

Review Report Form 1:

-Line 89: PEG is polyethylene glycol, please write it here, not mention it first in line 310.

Response: Thank you very much for your useful comments. Based on your suggestion, we have added polyethylene glycol instead of PEG, please see L89.

-Some mistypings occur, spaces are needed between numbers and dimensions between words, and after sentence ending dots. (line 93-95, 98, 100, 102, 103, 110-116, 123, 132, 137,

Response: Thank you very much for your useful comments. We have modified it. At the same time, we checked this kind of problem in the manuscript and corrected it. please see L94-115.

-In lines 127-129, and 133 equations would look better in form of Equation 1, Equation 2,

Response: Thank you very much for your comments, The equations appearing in the manuscript are represented in the form of equation 1 and equation 2, please see L127-128, L133-134.

-The already named equations should be formatted to look more uniformed

Response: Thank you for your kindly suggestion. We have revised the equations in the manuscript.

-Line 151-152, 162-163 Eq.(2) and Eq.(4) nomenclature should be revised and corrected e.g. using both forms Μ(X_j ) and μ(Xj) for the same factor is confusing for the readers.

Response: Thank you for your kindly suggestion. We have revised and corrected the nomenclature of equation (2) and equation (4) to use the same form for the same factor, please see L164-167.

-Line 137 the sentence should be rephrased since words seem to be missing from it, and it makes harder to understand it. Maybe, it can be divided in 2 separete sentences.

Response: Thank you for your kindly suggestion. We have divided in 2 separete sentences for Line 137 sentence, please see L139-141.

-Line 142 year is not needed in this form for the Xu et al. [18] reference.

Response: Thank you for your kindly suggestion. We have changed the description of this section, please see L144-146.

-Line 151-153: X_man should be X_max

Response: Thank you very much for your comments, We have change X_man to X_max, please see L164-166.

-Please, clarify CK. Is this the abbreviation for control? If yes, clarify it in the text (line 163).

Response: Thank you very much for your comments, the CK is control, We have modified it, please see L176.

-Line 168-170 is recommended to replace into Methodology

Response: Thank you very much for your comments, Redescription of the results section, with the description of the method removed, please see L181-183.

-In the Methodology section (probably after line 165), unfortunately, there is no part which presents the applied statistical analysis e.g. Cluster analysis mentioned in Results section (lines 183-184)

Response: In the Methodology section, We have added statistical analysis methods on cluster analysis, please see L179-180.

-strange comma character is used in lines 185-190, and lines 199-202, line 224 these should be corrected.

Response: Thank you very much for your comments, we have modified strange comma character, please see L198-202.

-For Fig. 1 part A measure scale bar would be needed.

Response: Thank you for your kindly suggestion. we have measure scale bar in Fig. 1 part A.

-Fig. 1 part B does not express the 3 clusters, but 2, I suggest to redo it in order to have a nice visual support.

Response: Thank you for your kindly suggestion. I've changed the image color to become 3 clusters.

-Table 1 Field names should be edited before publish

Response: Thank you for your kindly suggestion. we have edited table 1 Field names.

- Please correct 1 st to 1st, and 2 st to 2nd if you mean it this way.

Response: Thank you very much for your useful comments. We have modified 1 st to 1nd, and 2 st to 2nd At the same time, we checked this kind of problem in the manuscript and corrected it.

-Fig. 3 and 4: diagrams should be edited since it is hard to read the axe titles in this form, and Legend is not formatted the same, please redo those, without spacing errors.

Response: Thank you very much for your useful comments. we have edited Fig. 3 and 4 diagrams, enlarged the axe titles, pictures look clearer.

-Please, revise figure title text of Fig. 3.

Response: Thank you very much for your useful comments. we have revised figure title text of Fig. 3.

-Line 263: please correct „Chang- jing 616” to „ ’Changjing 616’ ”.

Response: Thank you very much for your useful comments. We have modified it, At the same time, we checked this kind of problem in the manuscript and corrected it. please see L275.

-Table 2 is not well edited and hence it is hard to use.

Response: Thank you very much for your useful comments. We have re-edited Table 2.

-Line 303: please, revise this Table title sentence.

Response: Thank you very much for your useful comments. We have revised this Table 2 title sentence.

 Best wishes, Sincerely yours, Xiaoshuang Wei E-mail: weixiaoshuang@jlau.edu.cn

Reviewer 2 Report

Thank you very much for your invitation to review the manuscript entitled [Physiological characterization of drought responses and Screening of rice under dry cultivation] (ID agronomy-2003428).

I suggest some improvements in my review below and in the attached Pdf version, I recommend the publication after major revision.

The author must follow the following comments:

Ø  Title: Physiological characterization of drought responses and Screening of rice varieties under dry cultivation

Ø  Line 32, Italic

Ø  Line 58, mesophyll

Ø  Line 59 CO2

Ø  Line 62   It is better to cite recent research such as: https://www.mdpi.com/2079-7737/10/6/520

Ø  Line 63   It is better to cite recent research such as :https://www.mdpi.com/2076-3921/10/3/398

Ø  https://www.mdpi.com/2223-7747/9/6/733

Ø  Line 89   Please write the full words and then the abbreviations in parentheses:  polyethylene glycol (PEG)

Ø  Line 117   You did not mention the soluble proteins in the methods; However, you mentioned them in the results, How????

Ø  Line 123   2nd

Ø  Line 134 You must add the methods in details and References for determination the following: net photosynthetic rate, stomatal conductance, transpiration rate, intercellular CO2, Proline, SOD, POD, CAT and Malondialdehyde

Ø  Line 145  Before this section, you should mention how and when did you determined the yield characteristics

Ø  Line 178 rewrite

Ø  Lines 211,213 2nd

Ø  Line 251 You did not mention the soluble proteins in the methods, but you mentioned them in the results, How????????

Ø  Line 258 You did not mention the soluble proteins in the methods

Ø  Line 270  CO2

Ø  Lines 274 you did not mention water use efficiency in the methods section

Ø  Line 288 you did not mention these characters in the methods

Ø  Line 348 periods

Ø  Lines 352, 353 (and)  should be changed to as well as

Ø  Line 360 Zhao et al. [34]

Ø  Line 395 CO2

Ø  The discussion needs improvement and you should discuss why MDA and Proline increased in tolerant varieties than non-tolerant one 

Ø  In the references section, there are a lot of mistakes with unfollow the journal rules

Ø  Please see the Pdf version

Best regards

Author Response

Point-by-point response to reviewers

Manuscript Title: Physiological characterization of drought responses and Screening of rice varieties under dry cultivation

Dear Editor:

First of all, we deeply appreciate the editor for your time and efforts. Based on your comments, we have uploaded the data in the manuscript and corrected the relevant experimental details. The revised parts are marked in different colors in the manuscript. Our specific response is as follows.

Review Report Form 2:

Ø Title: Physiological characterization of drought responses and Screening of rice varieties under dry cultivation

Response: Thank you very much for your useful comments. We have add ‘varieties’ in title.

Ø Line 32, Italic

Response: Thank you very much for your useful comments. We have modified it, please see L32.

Ø Line 58, mesophyll

Response: Thank you very much for your useful comments. We have modified it, please see L58.

Ø Line 59 CO2

Response: Thank you very much for your useful comments. We have modified it, At the same time, we checked this kind of problem in the manuscript and corrected it. please see L59.

Ø Line 62 It is better to cite recent research such as: https://www.mdpi.com/2079-7737/10/6/520

Response: Thank you very much for your useful comments. We quote from this recent literature, please see L62.

Ø Line 63 It is better to cite recent research such as :https://www.mdpi.com/2076-3921/10/3/398

Response: Thank you very much for your useful comments. We quote from this recent literature, please see L63.

Ø https://www.mdpi.com/2223-7747/9/6/733

Response: Thank you very much for your useful comments. We quote from this recent literature, please see L63.

Ø Line 89 Please write the full words and then the abbreviations in parentheses: polyethylene glycol (PEG)

Response: Thank you very much for your useful comments. we have write the full words for PEG, please see L89.

Ø Line 117 You did not mention the soluble proteins in the methods; However, you mentioned them in the results, How????

Response: Thank you very much for your useful comments. we have added soluble proteins in the methods, please see L150-154.

Ø Line 123 2nd

Response: Thank you very much for your useful comments. We have modified 2 st to 2nd, At the same time, we checked this kind of problem in the manuscript and corrected it.

Ø Line 134 You must add the methods in details and References for determination the following: net photosynthetic rate, stomatal conductance, transpiration rate, intercellular CO2, Proline, SOD, POD, CAT and Malondialdehyde

Response: We have modified this part of the material approach, please see L136-148.

Ø Line 145 Before this section, you should mention how and when did you determined the yield characteristics

Response: Thank you very much for your useful comments. we have added determined the yield characteristics, please see L155-157.

Ø Line 178 rewrite

Response: Thank you very much for your useful comments, Table 1 has been re-edited.

Ø Lines 211,213 2nd

Response: Thank you very much for your useful comments. We have modified 2 st to 2nd, At the same time, we checked this kind of problem in the manuscript and corrected it. please see L223-226.

Ø Line 251 You did not mention the soluble proteins in the methods, but you mentioned them in the results, How????????

Response: Thank you very much for your useful comments. we have added soluble proteins in the methods, please see L149-154.

Ø Line 258 You did not mention the soluble proteins in the methods

Response: Thank you very much for your useful comments. we have added soluble proteins in the methods, please see L149-154.

Ø Line 270 CO2

Response: Thank you very much for your useful comments. We have modified it, At the same time, we checked this kind of problem in the manuscript and corrected it. please see L282.

Ø Lines 274 you did not mention water use efficiency in the methods section

Response: Thank you very much for your useful comments. we have added water use efficiency in the methods, please see L140-141.

Ø Line 288 you did not mention these characters in the methods

Response: Thank you very much for your useful comments. we have added determined the yield characteristics, please see L155-157.

Ø Line 348 periods

Response: Thank you very much for your useful comments. we have added ‘periods’, please see L366.

Ø Lines 352, 353 (and) should be changed to as well asØ Line 360 Zhao et al. [34]

Response: We have already cited Zhao et al. article in these two parts, please see L378 and L381.

Ø Line 395 CO2

Response: Thank you very much for your useful comments. We have modified it, At the same time, we checked this kind of problem in the manuscript and corrected it. please see L419.

Ø The discussion needs improvement and you should discuss why MDA and Proline increased in tolerant varieties than non-tolerant one 

Response: Thank you very much for your useful comments. We have added to the discussion the increase in MDA and proline in tolerant varieties compared to non-tolerant varieties, please see L397-409.

Ø In the references section, there are a lot of mistakes with unfollow the journal rules

Response: Thank you very much for your useful comments. We have modified it, At the same time, we checked this kind of problem in the manuscript and corrected it. 

Best wishes, Sincerely yours, Xiaoshuang WeiE-mail: weixiaoshuang@jlau.edu.cn

Round 2

Reviewer 2 Report

Dear Editor

The manuscript is acceptable in the present form

Regards